# Ultrasonic Activated Biochar and Its Removal of Harmful Substances in Environment

**DOI:** 10.3390/microorganisms10081593

**Published:** 2022-08-08

**Authors:** Juanjuan Wang, Wenshu Li, Zhirui Zhao, Florence Sharon Nabukalu Musoke, Xiaoge Wu

**Affiliations:** 1Environment Science and Engineering College, Yangzhou University, Yangzhou 225009, China; 2Key Laboratory of Cultivated Land Quality Monitoring and Evaluation, Ministry of Agriculture and Rural Affairs, Yangzhou 225127, China; 3Hebei Province Key Laboratory of Sustained Utilization & Development of Water Recourse, School of Water Resources and Environment, Hebei GEO University, Shijiazhuang 050031, China; 4Jiangsu Provincial Laboratory of Water Environmental Protection Engineering, School of Environmental Science and Engineering, Yangzhou University, Yangzhou 225127, China

**Keywords:** ultrasound, biochar, cavitation, sonocatalyst, CO_2_ capture, soil remediation

## Abstract

Biochar has been widely used in the fields of environment and energy, and green preparation can make biochar-based materials more environmentally friendly. Particularly, in the low-temperature pyrolysis of biochar, labile C with low biological toxicity is the main influencing factor of bacteria in soil. Therefore, it is worth studying to develop the fabrication technology of low-temperature pyrolysis biochar with rich pore structure. The mechanical effect of ultrasonic cavitation is considered to be an effective strategy for the preparation of biochar. However, the sonochemical effects on biochar remain to be studied. In this review, ultrasonic modification and ultrasonic-chemical modification on biochar has been reviewed. Metal oxide/biochar composites can also be obtained by an ultrasonic-chemical method. It is worth mentioning that there have been some reports on the regeneration of biochar by ultrasound. In addition to ultrasonic preparation of biochar, ultrasound can also trigger the sonocatalytic performance and promote the adsorption ability of biochar for the removal of harmful substances. The catalytic mechanism of ultrasound/biochar needs to be further investigated. For application, biochar prepared by ultrasound has been used for the removal of heavy metals in water, the adsorption of carbon dioxide, and soil remediation.

## 1. Introduction

In the context of environmental protection, human society is committed to seeking new materials that are “green”, low-cost, efficient, and sustainable. Among them, carbon prepared from biomass has large reserves and is renewable. That makes it one of the most promising new materials. In fact, biomass resources are abundant. Biochar is usually prepared from plants that directly synthesize organic matter by photosynthesis—such as crops [1], rice husks [2], and waste wood [3]—but also from urban waste [4]. Efficient and comprehensive biomass utilization has been demonstrated in energy generation, ecological agriculture, environment restoration, and building materials [5]. When biomass is burned, it contains a lot of carbon and plant nutrients, which make it effective as a soil conditioner to improve soil quality [6]. In addition, biomass carbon has a pore structure with a high specific surface area and rich surface functional groups that adsorb heavy metals and organic pollutants from water [7].

Recently, biochar has demonstrated broad applicability in the field of energy and the environment. Its application in fuel cells has also been reported [8]. There are still some problems in biochar’s comprehensive application, however. Today’s preservation and transformation processes the waste of biomass resources. The worst is, of course, the burning of agricultural wastes such as straw in the field, but sawdust is discarded in bulk in forestry-product processing, and food processing routinely sends shells and skin to a landfill. Such practices pollute the environment and waste a considerable quantity of potentially useful biomass resources [9,10,11]. This makes finding a new method for preparing biochar and new large-scale uses for biomass an important research challenge. In today’s world, an important constraint is that any solvents used must be environmentally “friendly”.

Biochar is a solid formed through the thermochemical decomposition of biomass. It is a multifunctional material, which contains a lot of carbon and nutrients. It can be used as a soil conditioner to improve soil fertility and further improve crop yield [6]. In addition, biochar has a large specific surface area and contains more oxygen-containing active groups on the surface, which can adsorb heavy metals and organic pollutants in soil or sewage [1]. At present, biochar is usually produced through the anoxic high-temperature pyrolysis of biomass [12]. However, that thermal decomposition method has some shortcomings—slow heating, long reaction time, large energy consumption, low heat transfer efficiency, and uneven heating of the raw materials. This has led researchers to propose new methods.

One popular fabrication is hydrothermal carbonization. This is low-temperature thermal decomposition with a short reaction time and high yield. The waste biomass (carbohydrates with other organic molecules), a catalyst and water are heated to a temperature below 400 °C at elevated pressure. The elevated temperature and pressure accelerate the physicochemical interaction between the biomass and the solvent, which promotes the reaction between ions and acid or alkali. That decomposes the carbohydrate structure in the biomass, forming a carbonaceous precipitate. The aqueous medium of hydrothermal carbonization is conducive to the formation of oxygen-containing functional groups on the surface of materials in the carbonization process, so the carbonization products generally bear abundant surface functional groups.

Ultrasound is applicable to the preparation of carbonaceous materials from biomass. Ultrasonic preparation has been shown to efficiently prepare high-purity micro- and mesoporous and multistage porous carbonaceous products with a large specific surface area [13,14]. Applying ultrasound in carbonization and activation alleviates problems of low specific surface area and single pore structure. Indeed, ultrasound treatment can efficiently prepare material with a multistage pore structure and an adjustable micropore/mesopore ratio.

This review first introduces the effects of biochar on bacteria in soil. It then discusses in detail the preparation of biochar using ultrasound and the material’s resulting structural and catalytic properties. The latest research results will be reviewed, focusing on biochar’s application in water treatment, soil remediation, and new energy. A new biochar preparation method using microwave sonochemistry will be described. Finally, the obstacles to its implementation, directions for its further development and application, and some other challenges will be discussed. This report is intended to serve as a good guide to the rational use of biomass to prepare functional carbon materials and to improve their practical performance.

## 2. Effects of Biochar on Bacteria in Soil

### 2.1. Main Physical and Chemical Properties of Biochar

For the soil environment, the main physical and chemical properties of biochar include porosity, specific surface area, hydrophilicity/hydrophobicity, nutrient content, and pH. Generally speaking, the porosity and surface area of biochar usually increases with the increase in pyrolysis temperature. Appropriate porous structure can provide a good place for bacterial to grow. However, more importantly, biochar’s adsorption capacity for water and labile C should be considered [15]. Water and labile C provide essential nutrients for bacterial growth. The aromatization degree of biomass carbon is deepened, the hydrophobicity is enhanced, and the hydrophilicity is weakened. A review work pointed out that labile C in biochar is considered to be the main factor affecting bacterial growth. Interestingly, from the perspective of nutrients, the content of labile C will decrease with the increase in pyrolysis temperature [15]. This poses a challenge for the preparation of biomass carbon, whether we can prepare materials with high specific surface area at low pyrolysis temperature. In addition, not all labile C can be used as a carbon source for bacterial growth [16]. For example, xylenol in labile C is toxic to most bacteria and may stress the growth of bacteria. It is also worth noting that biochar is usually alkaline. In fact, for the same material, the higher the pyrolysis temperature, the higher the pH of biochar will increase. Alkaline biochar can bind hydrogen ions in soil solution, which may change soil acidification, leading to improved soil properties for microbes and plants [17].

### 2.2. Bacterial Response

The response of bacteria in soil to biochar is mainly reflected in the impact of biochar on bacterial diversity and community structure, which depends on the type of biochar, soil type, crop type, and planting time. For example, low-concentration carbon treatment promoted the plant height and biomass yield of rape, while high-concentration carbon treatment did not promote the plant height and biomass of rape. Biochar increased the relative abundance of Bacteroides in an acid purple soil [18]. Perhaps due to various influencing factors, the reports on the impact of biochar on microbial diversity are inconsistent [15]. In fact, in terms of response to bacteria in soil, meta-analysis is an important technical means that can reflect the impact of biochar on soil microbial enzyme activity. For example, meta-analysis showed that biochar significantly increased urease and alkaline phosphatase activities [19]. However, researchers have also observed the inhibition of biochar on urease activity. This may be the result of the oxidation reaction between free radicals produced by biochar and urease. Researchers believe that the effect of biochar on the activities of N and P enzymes depends on the type of biochar and their dosages [20]. Labile C provided by biochar may play an important role in this process.

Biochar can be used as a shelter for microorganisms because of its pore structure. Some bacterial cells can attach to the surface of biochar in a very short time [19]. Biochar can also provide nutrients for soil microorganisms through the adsorption of nutrient cations and inorganic anions with surface functional groups of biochar. In addition, biochar as a soil conditioner can reduce the toxicity of soil pollutants to soil microorganisms. Fixing soil pollutants on biochar, thereby reducing their bioavailability, may be the main reason for reducing the toxicity of soil pollutants to microorganisms and increasing microbial biomass. For example, the application of straw biochar can lead to an increase in the organic binding of heavy metals [21]. This can help bacteria survive in contaminated soil. However, some compounds in biochar are called microbial inhibitors, including carboxylic acids, ketones, furans, etc., which are usually considered as VOCs adsorbed on biochar [22]. In conclusion, physical and chemical properties of biochar will affect bacterial growth (Figure 1), while biochar’s promoting or inhibiting effect on bacteria is not very specific. This needs further study to evaluate the environmental benefits and the risks of biochar application.

### 2.3. The Effect of Biochar on the Nutrient Cycling of Soil by Acting on Bacteria

The physical and chemical properties of biochar play an important role in regulating the soil N cycle, changing the activity of nitrifiers and bacterial community composition, and affecting the soil nitrification process and N_2_O emissions [23,24]. Biochar can improve the availability of soil P by changing microbial communities, because it can provide suitable growth conditions for microorganisms. A microcosm experiment showed that the application of biochar increased soil orthophosphate and pyrophosphate, reduced the content of monoesters, and thus improved soil P components as well as P availability. However, P fixation is temporary and will be released again after microbial death [25].

At present, most studies have focused on the impact of biochar on bacterial communities in soil. However, in soil ecosystems, the cycles of C, N, and P usually occur together. Regulation such as biochar addition may disturb the coupling of microbial C, N, and P cycles and change the soil nutrient status. The microbial communities, dominant groups, and functions of biochar still need further investigation. From the perspective of bacterial growth, biochar prepared by high temperature can be used as a shelter for microorganisms because of its rich pore structure. However, in the low-temperature pyrolysis of biochar, labile C with low biological toxicity is the main influencing factor of bacteria. Therefore, it is worth studying to develop the fabrication technology of low-temperature pyrolysis biochar with rich pore structure.

## 3. Ultrasonic Modification of Biochar

Porous carbon materials have attracted a lot of research attention due to their large specific surface area, easily accessed active sites, and easy regulation of pore structure. They are widely used in energy storage, adsorption, water purification, and catalysis [26]. The preparation of porous carbon materials from biomass has been considered a green path to solving the pollution problems posed by waste biomass. In the preparation process and when the product serves as a catalyst, cavitation can achieve better mass transfer, eliminate any uneven concentration at the microscale interfaces, accelerate the reactions, and inhibit particle agglomeration. It has been reported that ultrasound can be used to prepare biochar-based materials [27]. It is therefore of great scientific and economic significance to understand the mechanisms involved in preparing porous carbon materials from biochar using ultrasound. That could allow improving the porous carbon’s catalytic and adsorption properties.

### 3.1. Ultrasound Modification

Equipment, the chemical environment, and parameters such as frequency and intensity have critical impacts on reactions promoted using ultrasound. However, researchers have not yet systematically investigated the effects of ultrasound frequency or intensity in biochar modification. This review summarizes the optimization work to date seeking better ultrasonic treatment conditions, which should help to improve biochar modification processing using ultrasound. The ultrasonic modification on biochar is demonstrated in Figure 2.

Table 1 shows the work to date on ultrasonic modification of biochar. Most of it used an ultrasound probe, but some used a low-frequency ultrasound bath. Low frequencies at high power have been favored. The reported frequencies have ranged from 20 to 170 kHz, and the power from 20 to 700 W. The reports point out that the cavitation induced by high-intensity ultrasound can cause exfoliation, which will significantly impact the porosity and physical adsorption properties of biochar.

Cavitation corrosion can increase the surface area of biochar. Aneeshma’s group used 40 KHz, 1000 W ultrasound and 170 kHz, 1000 W ultrasound to treat biochar from mixed softwoods as a pretreatment before pyrolysis [28]. They found that ultrasonic pretreatment developed a more porous structure in the material and increased the biochar’s specific surface area. It is worth noting that in that work, 170 kHz ultrasound was significantly more effective than 40 kHz ultrasound in improving the biochar’s Brunauer, Emmett, and Teller (BET) surface area. Experiments to further optimize the conditions are needed. 

A group led by Sajjadi has reported that sonication times longer than 20 s did not further enhance the surface area of biochar in their experiments [41]. Thus, the traditional influencing factors of pyrolysis mainly include carbonization temperature and carbonization time, which have been investigated [42]. In terms of ultrasonication time, most of the prior work experimented with relatively short treatments, possibly due to the use of powerful ultrasound. For example, after pyrolysis, pinewood biomass was treated at 20 kHz and 700 W intensity for 30 or 60 s. Even in such a short time, the porosity was enhanced [30]. Biochar derived from sewage sludge was treated at 24 kHz and 400 W for 30 s, resulting in increased pore volume and surface area [31]. Thus, even ultrasonication for 30–60 s can effectively improve the surface area and even open holes on the surface of carbon materials [30,31,33,43]. Shorter treatment time is of course an advantage in commercial application of ultrasound.

### 3.2. Ultrasonic-Chemical Modification

There have been experiments mixing biochar deionized water and chemicals before ultrasonication [31,41,44]. The urea can also help to generate an appropriate pore-size distribution and high specific surface area. This needs to be further studied (Figure 3).

Only some research reports emphasize temperature control. For example, when an ultrasonic cleaner was used to modify rice husk biochar, the temperature was controlled at 30 °C during 1.5 h of ultrasonication [2]. The atmosphere and temperature are important treatment parameters. Higher temperatures can even inhibit some ultrasonic chemical reactions. Work systematically exploring the role of atmosphere and temperature in ultrasonic modification of biochar is badly needed.

Ultrasonic modification positively affects the specific surface area of carbon from biomass, but it has no noticeable effect on the elemental composition or on the oxygen-containing functional groups on the biochar’s surface. The principle of chemical modification is to chemically change the functional groups on the material’s surface. The main methods are acid and alkali modification. Acid modification enhances the hydrophilicity of biomass carbon by increasing the surface concentration of acidic groups. Alkali modification forms a positive charge on the surface which promotes the adsorption of negatively-charged substances.

The combination of ultrasound and chemical modification can often better improve the performance of biochar. For example, combining 20 kHz ultrasound with urea in biochar activation was found to enhance the adsorption of heavy metals [41]. Biochar fabricated from camphor leaves was sonicated with NaOH solution for 30 min and the combined treatment resulted in more surface groups, larger surface area, and greater pore volume, leading to higher sorption capacity for heavy metals [45]. In addition, a group led by Zhang reports [46] that imidacloprid adsorption in aqueous solution on carbon from corn cob treated with KOH was enhanced compared with the original carbon. Aswani has similarly reported that ultrasound acid treatment improved the biosorption capacity of *Merremia vitifolia* biomass [47]. Alternatively, ultrasound activation could be followed by a chemical modification. Either EDC−HOBt or KOH improves the CO_2_ adsorption [34]. A group led by Bispo has reported that using K_3_PO_4_ and sonication, biochar formation is reduced while gas yield is enhanced [48]. When biochar from waste tea feedstock was mixed with 0.3 M citric acid and ultrasonically treated for 2 h the pore volume and pore size were increased, leading to thermostability and high Hg0 removal (approximately 98.6%). The sonication also improved the char’s surface morphology, thermal stability, and regeneration ability [49]. Various solvents have been applied in the ultrasonic exfoliation of biochar [50]. To obtain a small number of stacked layers from such exfoliation requires careful control of the solvent’s properties, and biochar-solvent intermolecular interactions.

When biochar from guava seeds was mixed with Ch_2_Cl_2_, KOH, or H_3_PO_4_ before 30 min of sonication, CH_2_Cl_2_ enhanced the hydrolytic activity from 190 to 258 μmol g^−1^ min^−1^ while the other two treatments reduced it [51].

To summarize, ultrasound combined with different chemicals can controllably adjust biomass carbon’s physical and chemical properties.

### 3.3. Metal Oxide-Biochar Composites

Combining ultrasound with acid treatment can generate a biochar with magnetization saturation, which more quickly leaches NiCl_2_ and ZnCl_2_. In that process, pyrolyzed Sapelli biochar was mixed with 0.1 M HCl and sonicated for 15 min at 20 kHz and 600 W [52]. Rice husk biochar modified with FeCl_3_ solution and 1 h of sonication can be used for sludge dewatering [2]. Ultrasound-assisted in situ precipitation of Bi_4_O_5_Br_2_ has been used to prepare a biochar/Bi_4_O_5_Br_2_ photocatalyst [53]. Using sonication, CaCO_3_ nanoscale particles have been grown on tragacanth gum biochar for Pb^2+^ removal [54]. Furthermore, 40 min of sonication has been shown to enhance the S_BET_ area and pore volume of metal oxide/biochar complexes. Luo’s group used 0.5 h of sonication to precipitate zinc sulfide on peach wood activated carbon, which then showed remarkable photocatalytic ability [55]. Magnetic Fe_3_O_4_ particles have been loaded on biochar, and the composite has used an adsorbent in removing antibiotics from aqueous solution [56].

There is also sonocrystalization. Ultrasound is used in preparing crystalline TiO_2_/lignocellulosic carbon for photocatalytic reactions [57]. A surfactant-free biochar bearing TiO_2_ can be formed in an hour in a 35 kHz ultrasound bath at 560 W. The composite can photocatalytically degrade 64.1% of the phenol in a liquid medium, and selectively oxidize 90% of methanol in a gas phase reaction [37]. However, in fact, a 40 kHz bath operating at only 50 W is capable of loading TiO_2_ nano-scale particles onto the surface of porous biochar [58].

### 3.4. Microwave-Ultrasound Fabrication of Biochar

Microwaves heat by generating high-frequency reciprocating motion of polar molecules in the heated body. Colliding molecules generate heat through friction, raising the temperature of the internal and external parts of the body rapidly and evenly. Microwave heating has been applied in the preparation of biochar [13]. Sonication helps the doping of biochar with ZnS, while microwave heating is vital for giving the composite photocatalytic activity [55]. Magnetic biochar is first activated using microwaves before the activated biochar is sonicated for 40 min to reduce the particle size and disperse the magnetic iron oxide [59]. Combining acid or base treatment with ultrasound and microwave irradiation can remarkably enhance the surface area of biochar [60].

### 3.5. Ultrasonic Regeneration

The inactivation of a catalyst or adsorbent is a bottleneck in removing pollutants. Fast and complete regeneration is important. At present, regeneration usually involves either calcination, cleaning, oxidation-reduction, or resin adsorption. However, a group led by Ma has reported the ultrasonic renewal of biochar. They combined ultrasound with ethanol to regenerate magnetic biochar sludge. The first regeneration cycle reached 90.3% regeneration [4]. In other work (Figure 4a), hydroxyl-activated magnetic biochar from sugarcane bagasse was regenerated using ultrasound and ethanol [61]. Ultrasonic cavitation and ethanol extraction restored a stable adsorption capacity of 180 mg/g after five reuse cycles, which was close to the capacity of the fresh biochar. More than 99% of the adsorption capacity of magnetic microporous biochar from loofah sponge was similarly retained (Figure 4b) after five reuse cycles [62]. Effective regeneration is important, but relatively little is published on the subject, perhaps because of commercial considerations. In addition, the mechanism of ultrasonic regeneration has not been fully explored.

## 4. Pollutant Removal Using Ultrasound with Biochar

### 4.1. Heterogeneous Reactions

Heterogeneous sonocatalysis usually involves a solid catalyst sonoactivated through cavitation. Shear force, microjetting and free radicals all degrade organic contaminants, and solid particles of a catalyst such as Al_2_O_3_, ZnO, TiO_2_, or a carbonaceous material can help [19]. It is believed that particles reduce the cavitation threshold [19].

Biochar is a catalyst with good biocompatibility, which is inexpensive and environmentally friendly. It can play a significant role in environmental remediation [14,63]. The porous and irregular surface of biochar may enhance catalytic processes [64,65,66]. The addition of sludge biochar [67], pine wood-based biochar [68], and agroindustrial biochar [69] have all been shown to enhance the generation of free radicals better than sonication alone.

There are inorganic sonosensitizers that promote the production of free radicals under ultrasonication. Research in this area emphasizes regulating the physical and chemical structure of sonosensitizers (Table 2). 

Some of it aims to mediate hole–electron separation and generate free radicals. In a sound field, a sound-sensitive agent with a gaseous core and semiconductor properties generates carriers [79,80,81,82]. The carriers are separated and diffuse to the surface of the sound-sensitive agent. O_2_ captures the electrons in its conduction band to generate copious O^2−^. The holes of the valence band may generate OH^−^ in water, and some O^2−^ can also be reduced to OH^−^ [79,80,81,82,83] through electron induction.

Controlling the chemical composition of semiconductor materials and structural features such as ultrathin sheet structures or defects can substantially improve dynamic acoustic effects. Biomass has been used as a carrier for TiO_2_ in removing algae by applying dynamic acoustic effects [58]. The results show that supplementing biochar with iron can significantly reduce the sound pressure threshold, couple chemical dynamics, promote the formation of free radicals, and enhance algae removal.

Another research objective has been to improve the free radical yields in Fenton and Fenton-like reactions. A sound-sensitive agent with Fenton catalytic activity can enhance the production of free radicals. For example, sludge biochar can catalyze persulfate in a sound field to produce SO_4_^2−^ and OH^−^ radicals [68]. The species cycling of the iron in iron-doped biochar gives the catalyst a Fenton catalytic functionality, catalyzing H_2_O_2_^’^s production of reactive oxygen species [71]. In a sound field, rice husk biochar with MnO_2_ catalyzed the generation of H_2_O_2_ and degraded 100%.

A catalyst with nanoscale channels can also improve the reaction rate of free radicals. Qu has pointed out that the mass-transfer distance between the oxidant and the target pollutant in a confined space is very short. The resulting rapid mass transfer can promote the generation and accumulation of free radicals, which will then quickly collide with the target pollutant molecules, oxidizing them [84]. Biochar has abundant pores and surface functional groups, which can adsorb pollutants to the catalyst surface and react with contaminants in the tiny pores.

Free-radical mass transfer over short distances is the essential theoretical basis for regulating advanced oxidation reactions. The half-life of free radicals in water is about 10^−6^ to 10^−9^ seconds, so the limiting mass-transfer distance under ideal conditions is about 90 nm [85]. Free radicals’ short lifetimes do not allow for much diffusion. For advanced oxidation reactions involving biochar and ultrasound, it is important to understand the free radicals’ generation and mass transfer at the surface of the ultrasound generator. Will the generation and accumulation of free radicals be sufficient to ensure that free radicals will collide with pollutant molecules effectively? Can the sound generation parameters effectively regulate free radicals’ formation and mass transfer?

### 4.2. Ultrasound-Assisted Adsorption

Although ultrasound is often used to enhance desorption, it can also improve the adsorption of organic pollutants. Adsorption of reactive yellow dye 145 onto biochar from marine *Chlorella* sp. promoted by 35 kHz ultrasound allowed 99% removal of solid waste within 1 min [86]. Ultrasound-assisted adsorption has also been reported using porous biochar from almond shells, and it removed 96.88% of sulfamethoxazole [86].

The adsorption mechanism has been investigated using FTIR spectra. They showed that the peaks of functional groups such as OH, C=C, C-OH, and C-H were enhanced after sonication. Sonication may thus help fix pollutant molecules inside the adsorbent’s pores. Biochar bearing Fe_2_O_3_ and activated with ultrasound actively adsorbs salicylic acid and ketoprofen [87]. Diao reports that ultrasonic adsorption is the first step in Pb (II) and phenol removal using sludge biochar and ultrasound in precipitation, reduction, and Fenton-like oxidation [67]. Although there have been only limited experimental results showing that ultrasound can enhance adsorption on biochar, the potential mechanism is that ultrasound may drive organic molecules to the activated carbon. In any case, ultrasound should effectively prevent agglomeration of the adsorbent, improving the contact area and increasing the adsorption capacity.

## 5. Discussion

Biochar has a wide range of sources, and the successful preparation of functional materials is conducive to alleviate environmental problems. At present, some researchers have proposed adding biochar to soil, which can improve soil quality. However, several reports also pointed out that the addition of biochar may have negative effects. In this paper, the effect of biochar on soil bacteria was investigated from the perspective of bacteria in soil. We found that in some reports, biochar can promote the growth of bacteria, and then promote the degradation of organic matter, heavy metal passivation, and other beneficial processes. However, some studies demonstrated that biochar also inhibits bacterial activity. In general, it might be concluded as biochar prepared at low temperature contains labile C, and some components in this mixture can become a carbon source for bacterial growth. However, when the preparation temperature increases, the content of labile C decreases. Then, such findings do not guide us to determine the existing biochar preparation process as low-temperature preparation. Because the main advantages of biochar are considerable specific surface area and abundant pore structure, which could be used as sites for adsorption and/or catalytic reactions. How to prepare biochar with high relative area at low temperature is an interesting direction of biochar production in the future.

The report of ultrasonic preparation of porous carbon materials is not rare. Relatively speaking, the temperature required for ultrasonic treatment can be slightly reduced. This makes it possible to obtain biomass carbon with high specific surface area and rich in labile C. However, during laboratory preparation, due to the varied parameters of sonochemistry, if it is not accurately controlled, the preparation results may not be repeatable. Therefore, in this paper, we reviewed the preparation of biochar by ultrasound, focusing on the parameters related to sonochemistry such as ultrasonic frequency and intensity. In addition to the physical and chemical effects of ultrasonic cavitation, we also introduced ultrasonic coupled chemical treatment, and ultrasound combined microwave treatment, which are potential and effective for low-temperature biochar fabrication. In order to better interpret the effect of ultrasound on biochar, we also investigated the regeneration and catalytic mechanism of ultrasound on biochar.

In future, environmental scientists and materials scientists should better communicate, focus on how to retain the labile C of biochar, and try to develop the specific surface area of materials. In addition, it is worth noting that labile C also contains some toxic VOCs. Therefore, the regulation of labile C component is also a challenge to the biochar preparation process.

## 6. Conclusions and Recommendations

Large-scale use of biomass is in many ways is a difficult problem. However, the work reported in this review suggests that using it as biochar has many advantages. In particular, labile C with low biological toxicity from biochar is the main influencing factor of bacteria in soil, which may enhance the property of soil for plant growth, but the content of labile C will be reduced under high-temperature pyrolysis. Therefore, it is worth studying to develop the fabrication technology of low-temperature pyrolysis biochar with rich pore structure. In this regard, ultrasound modification can make many biochar applications more attractive.

In the preparation process of biochar, cavitation can achieve better mass transfer, eliminate any uneven concentration at the microscale interfaces, accelerate the reactions, and inhibit particle agglomeration. Combining acid or base treatment with ultrasound and microwave irradiation can remarkably enhance the surface area of biochar, which may not require high-temperature pyrolysis. In addition, if larger-scale equipment were available, experiments using low-frequency sonication at high intensity would help determine whether the promising results from ultrasonic modification of biochar in the laboratory can be scaled up to pilot and industrial scales.

Biochar-based catalysts activated with ultrasound have also shown promise for contaminant removal, but most of the work has used low-frequency, high-intensity ultrasound. Biochar’s adsorption has been amply demonstrated, but its catalytic mechanisms call for further study. In particular, there may be some sonosensitive functional groups such as porphyrins on the surface of biochar, which react with oxygen or water to form free radicals under the action of ultrasound. This may provide a new idea for biochar modification and application.

The application of biochar in agriculture and soil remediation is receiving scholarly attention, but more work is needed on which characteristics of biochar have the greatest impact on soil chemistry and biology. Future research should experiment with biochar with different physical and chemical properties developed through ultrasonic treatment, and explore the impact of biochar properties on the physical and chemical properties of soil. Such research should gradually reveal the function mechanisms involved, providing a scientific basis for advances in agriculture and ecology management.

Further work is also required to determine the benefits of sonication in conjunction with other current treatments such as microwave treatment, plasma treatment or gas activation.

## Figures and Tables

**Figure 1 microorganisms-10-01593-f001:**
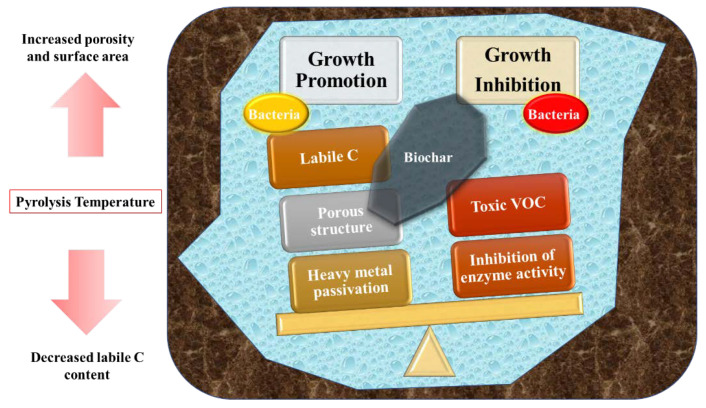
Schematic diagram of physical and chemical properties of biochar on bacterial growth.

**Figure 2 microorganisms-10-01593-f002:**
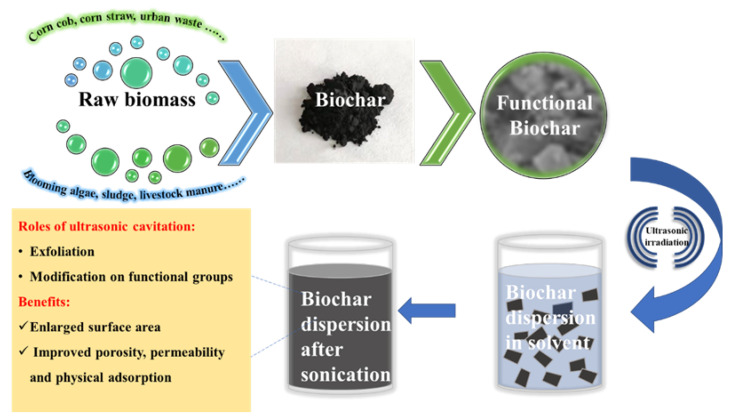
Schematic overview of ultrasonic modification on biochar.

**Figure 3 microorganisms-10-01593-f003:**
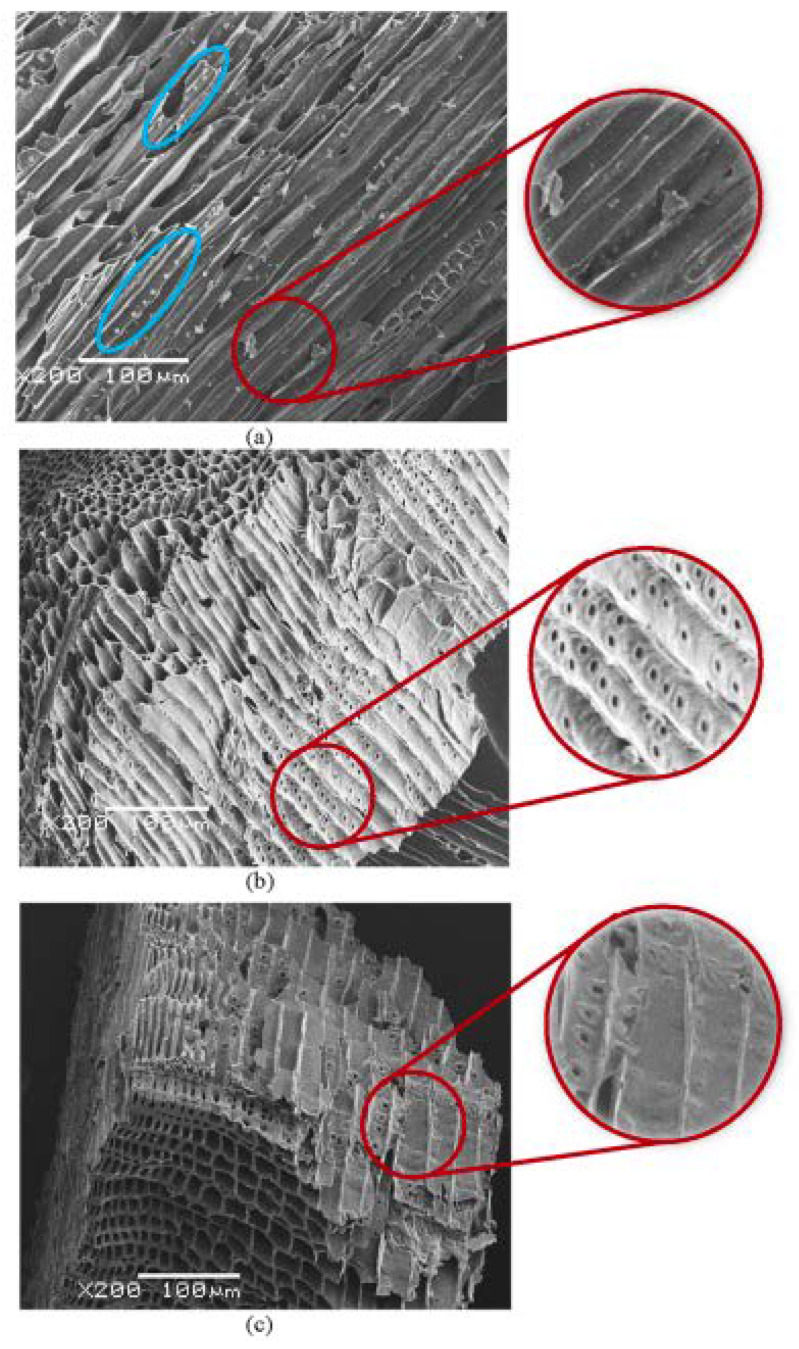
(**a**–**c**). SEM images of raw biochar (**a**), and biochar treated by ultrasound (**b**) and treated by urea/ultrasound (**c**). Reprinted with permission from ref. [41]. Copyright 2019 Elsevier B. V.

**Figure 4 microorganisms-10-01593-f004:**
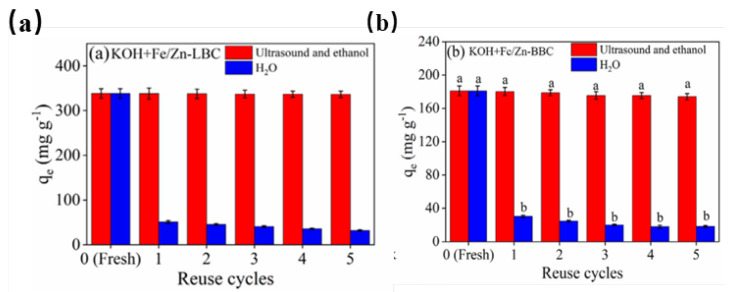
(**a**) Ultrasonic regeneration on Fe/Zn-loaded microporous loofah sponge biochar. Reprinted with permission from ref. [61]. Copyright 2021 Elsevier B. V. (**b**) Ultrasonic regeneration on Fe/Zn-loaded sugarcane bagasse biochar. Reprinted with permission from ref. [62]. Copyright 2021 Elsevier B. V.

**Table 1 microorganisms-10-01593-t001:** Effects of ultrasound parameters in biochar modification.

Material	Ultrasonic Conditions				Biochar Properties after Ultrasound Modification	Ref.
	Device	Frequency	Intensity	Time (s)		
**Biomass from mixed softwoods**	**Bath**	40 kHz 170 kHz	250 W 1000 W	3600 3600	Obtained a porous structure and increased heterogeneity of the surface	[27]
Woodchips	Bath	40 kHz 170 kHz	250 W 1000 W	3600 7200	A better surface morphology	[28]
Woodchips	Bath	40 kHz 170 kHz	250 W 1000 W	3600 7200 s	Enhanced surface area	[29]
Pine wood	Probe	20 kHz	700 W	30,60	Enhanced porosity	[30]
Sludge-derived biochar	Probe	24 kHz	400 W	30	Enhanced pore volume and surface area	[31]
Pine wood	Probe	20 kHz	475 W 700 W	30,60,180	Creating empty pores	[32]
Pine wood	Probe	20 kHz	700 W	30	A smooth surface with new circular pores	[33]
Pine wood-based biochar	Probe	20 kHz	700 W	30	Elevated adsorption capacity	[34]
*Caragana* *korshinskii*	Bath	45, 80, 100 kHz	300 W and 700 W	1800−14,640	Removed the ash content from the biochar and increased the specific surface area	[35]
Corn stover	Probe	20 kHz	500 W	60 s	Obtained multilayered and porous structures	[36]
Water bamboo husks	Probe	20 kHz	65 W	30−480	Improved the surface properties	[3]
Biochar	Bath	35 kHz	560 W	3600	Enhanced BET surface area	[37]
Biochar prepared from spent malt rootlets	Probe	20 kHz	4.32 W	n.a.	Surface activation	[38]
Milled miscanthus particles	Bath	40 kHz	300 W	3600	Synthesis of graphene oxide	[39]
Biochar	n.a.	20 kHz	475 W	300− 21,600	Exfoliation and enhanced reactivity of the surface functional groups	[40]

n.a.: Not available; BET: Brunauer, Emmett, and Teller.

**Table 2 microorganisms-10-01593-t002:** Biochar-based sonocatalysts for contaminant removal.

Biochar-Based Material	Contaminants	Ultrasonic Conditions				Results	Ref.
		Device	Frequency	Intensity	Time		
**1.5 g/L biochar from sludge**	**80 mL of 40 mg/L Pb (II) and/or 5 mg/L phenol solution**	Probe	20 kHz	50 W	60 min	98.9% of Pb (II) and 94.45% of phenol was removed.	[67]
90 mg/L pine wood-based biochar	250 μg/L sulfamethoxazole	n.a.	20 kHz	n.a.	30 min	100% of sulfamethoxazole was degraded (250 mg/L persulfate).	[68]
125 mg/L agroindustrial biochar	200 mL of 1 mg/L propylparaben solution	Probe	20 kHz	20–60 W/L	45 min	80% of propylparaben was degraded.	[69]
2 g/L Fe^0^ and Al^0^@sludge biochar	60 mL of 20 mg/L bisphenol A solution	Probe	n.a.	60 W	80 min	98.6% of bisphenol A was degraded [PS]0=3 mM	[70]
90 mg/L biochar	500 µg/L	Probe	20 kHz	36 W/L	120 min	90% trimethoprim (500 mg/L persulfate).	[38]
0.7 g/L MnFe_2_O_4_ and biochar derived from polar wood powder	200 mL of 20.0 mg/L methylene blue solution	n.a.	40 kHz	665 W	20 min	95% of methylene blue was degraded (pH=5, 15 mol/L H_2_O_2_).	[71]
0.5 g/L MnO_2_ with rice husk biochar	200 mL of 100 µM bisphenol A solution	Probe	20 kHz	130 W at 40% amplitude	120 min	100% of bisphenol A was degraded. [H_2_O_2_]_0_ = 10 mM	[72]
2 g/L magnetic biochar derived from food waste	10 mL of 50 mg/L methylene blue solution 10 mL of 50 mg/L methyl orange solution	Bath	37 kHz	35.3 W/L	60 min, 180 min	methylene blue and methyl orange 100% degraded (200 mM H_2_O_2_).	[73]
1 g/L magnetic biochar from rice bran	200 mL of 0.1 mM bisphenol A	Probe	20 kHz	51.95 W/L	40 min	94.25% of bisphenol A was degraded (10 mM H_2_O_2_).	[74]
sodium alginate-coated iron granules with biochar	100 mL of 100 mg/L ibuprofen	Bath	40 kHz	250 W	8 h	74.72% of ibuprofen was degraded.	[75]
50 mg/L TiO_2_ loaded on biochar	20 mL of 1.3 × 10^7^ cells per mL *Microcystis aeruginosa* cells	Bath	600 kHz	0.3 W/mL	90 s	the number of cyanobacteria cells decreased to 0.8 × 10^5^ cells per mL.	[58]
0.6 g/L ZnCr and LDH biochar	15 mg/L rifampicin	Bath	36 kHz	150 W	40min	100% of rifampicin was degraded with ultrasound and visible light irradiation.	[76]
1 g/L CeO_2_ on biochar	100 mL of 10 mg/L Reactive Red 84	n.a.	n.a.	450 W	60 min	98.5% of Reactive Red 84 was degraded.	[77]
43 mg/L *Tamarix hispida* biochar modified by lanthanum chloride	50 mL of 86 mg/L phenol	Bath	370 kHz	n.a.	63 min	99.43 % of phenol was degraded. (86 mg/L persulfate)	[78]

n.a.: Not available.

## Data Availability

Not applicable.

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
