# Peer review of "Ultrasonic Activated Biochar and Its Removal of Harmful Substances in Environment"

_microorganisms, 2022, doi:10.3390/microorganisms10081593_

Round 1

Reviewer 1 Report

Dear Authors,

many thanks for this very helpful review paper.

I got a very good impression about the state of research and the missing bricks.

From my point of view only very small adaptions suggested:

L81/82: improve their

L102: leading to improved

L252: CH2Cl2

L356: adsorption

Table 2: ckeck th printing style according to bolt and underlining + 6th example: poplar wood

L377: However, several reports

Congratulations and kind regards

Author Response

Thank you very much for the comments! We have revised the manuscript according to your comments.

L81/82: improve their

Response: We changed the sentence from “Appropriate porous structure can be a place for bacteria to improve their growth.” to “Appropriate porous structure can provide a good place for bacterial to grow.”

L102: leading to improved

Response: We changed “leading to an improved” to “leading an improved” as suggested.

L252: CH2Cl2

Response: We changed “Ch2Cl2” to “CH2Cl2” as suggested.

L356: adsorption

Response: We changed “Adsoption” to “adsorption”

Table 2: ckeck the printing style according to bolt and underlining + 6th example: poplar wood

Response: We modified the style of the table as suggested.

L377: However, several reports

Response: We added “several” before “reports” as suggested.

Reviewer 2 Report

This is the review of the manuscript entitled "Ultrasonic activated biochar and its removal of harmful substances in environment". The authors present an topic being in line with the mission of the Microorganisms. In general, this manuscript is well organized.

I have suggestion: The manuscript should contain the properties of biochars.

Reviewer 3 Report

1. I miss an index summarizing the different contents of the review.

2.Some examples of typical biochar with structures and characterizations are suggested to be provided.

3.Please use same time unit in Table 1.

4.”c” was miss in Figure 3.

5. Go through the manuscript and revise some spelling mistakes.
